# Innovative Alignment-Based Method for Antiviral Peptide Prediction

**DOI:** 10.3390/antibiotics13080768

**Published:** 2024-08-14

**Authors:** Daniela de Llano García, Yovani Marrero-Ponce, Guillermin Agüero-Chapin, Francesc J. Ferri, Agostinho Antunes, Felix Martinez-Rios, Hortensia Rodríguez

**Affiliations:** 1School of Chemical Sciences and Engineering, Yachay Tech University, Hda. San José s/n y Proyecto Yachay, Urcuquí 100119, Imbabura, Ecuador; garcia.dellano@yachaytech.edu.ec (D.d.L.G.); hmrodriguez@yachaytech.edu.ec (H.R.); 2Universidad San Francisco de Quito (USFQ), Grupo de Medicina Molecular y Traslacional (MeM&T), Colegio de Ciencias de la Salud (COCSA), Escuela de Medicina, Edificio de Especialidades Médicas, Instituto de Simulación Computacional (ISC-USFQ), Diego de Robles y vía Interoceánica, Quito 170157, Pichincha, Ecuador; 3Facultad de Ingeniería, Universidad Panamericana, Augusto Rodin 498, Benito Juárez 03920, Ciudad de México, Mexico; felix.martinez@up.edu.mx; 4Computer Science Department, Universitat de València, 46100 Valencia, Burjassot, Spain; francesc.ferri@uv.es; 5CIIMAR—Centro Interdisciplinar de Investigação Marinha e Ambiental, Universidade do Porto, Terminal de Cruzeiros do Porto de Leixões, Av. General Norton de Matos, s/n, 4450-208 Porto, Portugal; aantunes@ciimar.up.pt; 6Departamento de Biologia, Faculdade de Ciências, Universidade do Porto, Rua do Campo Alegre, 4169-007 Porto, Portugal

**Keywords:** antiviral peptide, multi-query similarity search, machine learning, StarPep toolbox, antiviral peptide dataset

## Abstract

Antiviral peptides (AVPs) represent a promising strategy for addressing the global challenges of viral infections and their growing resistances to traditional drugs. Lab-based AVP discovery methods are resource-intensive, highlighting the need for efficient computational alternatives. In this study, we developed five non-trained but supervised multi-query similarity search models (MQSSMs) integrated into the StarPep toolbox. Rigorous testing and validation across diverse AVP datasets confirmed the models’ robustness and reliability. The top-performing model, M13+, demonstrated impressive results, with an accuracy of 0.969 and a Matthew’s correlation coefficient of 0.71. To assess their competitiveness, the top five models were benchmarked against 14 publicly available machine-learning and deep-learning AVP predictors. The MQSSMs outperformed these predictors, highlighting their efficiency in terms of resource demand and public accessibility. Another significant achievement of this study is the creation of the most comprehensive dataset of antiviral sequences to date. In general, these results suggest that MQSSMs are promissory tools to develop good alignment-based models that can be successfully applied in the screening of large datasets for new AVP discovery.

## 1. Introduction

Antiviral peptides (AVPs) represent a promising, innovative, and unconventional approach in the ongoing fight against viral infections, a persistent global health concern [1]. The ever-present threat of viral outbreaks underscores the critical need for effective antiviral treatments. AVPs have emerged as a promising alternative because of their unique ability to target various stages of viral infections [2].

Viruses, notorious for their vast genetic diversity and adept replication within host cells, present immense challenges in disease containment [3]. Traditional antiviral drugs, although effective in certain instances, often have limited reach and may become obsolete because of resistance [4]. This treatment void has catalysed research into AVPs as alternative antiviral agents. AVPs have shown efficacy against numerous viruses, including lethal ones, like HIV [5], influenza [6], hepatitis [7], and emerging zoonotic viruses, such as Ebola [8] and Zika [9].

The therapeutic allure of AVPs is rooted in their distinctive mechanisms of action. They hinder viral attachment, fusion, and replication, offering a comprehensive strategy against viral infections [10]. Derived from synthetic libraries or sections of natural proteins, these peptides boast characteristics vital for their antimicrobial actions. Their low toxicity, high specificity, and efficiency mark them as potent contenders in medical applications [11].

Discovering and validating AVPs through lab experiments is resource-intensive, but computational strategies are gaining traction for their potential in identifying peptides with antimicrobial properties. Crafting robust computational models is imperative for effectively pinpointing potential AVPs. Currently, the strategy for processing expansive peptide databanks relies on machine learning (ML), enabling in-depth multidimensional data analysis. Conventional ML algorithms, such as SVM [12,13,14], kNN [15], RF [16,17,18], NN [19], and deep-learning (DL) algorithms [20,21,22], have shown prowess in discerning patterns in peptide sequences and assessing new ones.

However, a point of contention in this domain is the efficacy of DL models in forecasting AVPs. Many DL techniques necessitate vast experimentally validated peptide sequence datasets, which are often lacking. One remedy is “data augmentation”, although it is mostly untapped [21]. As García-Jacas et al. highlighted [23], DL does not significantly outpace traditional ML, and their algorithmic outputs frequently intersect. Additional challenges include the overrepresentation of specific sequences and imbalanced data, which can distort evaluations based purely on accuracy. Moreover, reproducibility challenges persist, as not all scientists disclose their source code or datasets, obstructing these methods’ widespread acceptance. Therefore, although ML-centric methods hold promise in predicting active peptides, their refinement is an ongoing endeavour [24,25].

Recent studies have showcased a non-trained supervised technique, the multi-query similarity search (MQSS), for predicting peptide bioactivities, including haemolysis [26], tumour-homing [27], and antiparasitic [28] activities, with impressive results. This method trumps conventional ML methods in several ways: it is user friendly, does not rely on web server availability, consumes fewer computational resources, and processes sequences with non-standard amino acids or varying lengths. Remarkably, MQSS models (MQSSMs) function without extensive training, relying instead on fine-tuning certain parameters, like the sequence alignment type and the similarity cutoff value. They can be developed without needing a negative dataset, a significant advantage given the scarcity of validated negative sequences, ensuring the learning phase is not skewed by data imbalances.

In prior research, we explored the AVP chemical space using interactive mining and complex networks via the StarPep toolbox (https://github.com/Grupo-Medicina-Molecular-y-Traslacional/StarPep (v0.8.5, accessed on 17 April 2023), [29]). We generated, disclosed, and made public diverse reducts of the AVP chemical space by implementing scaffold extraction in half-space proximal networks formed from the predefined chemical space. These scaffolds now serve as a foundation for designing reference/query datasets for MQSSM creation [30]. As the MQSS technique has not been applied to AVPs, our primary objective is to create MQSSMs for AVPs that match or surpass existing predictors while addressing the shortcomings of ML predictors. The MQSSMs were crafted using the available resources in StarPep toolbox. Our overarching goal is to enhance the methodologies for identifying and advancing AVPs, potentially revolutionising antiviral therapies. Additionally, we have assembled the most extensive dataset of antiviral sequences to date, essential for developing effective AVP predictors and ensuring robust model validation.

## 2. Data and Methods

### 2.1. The Multi-Query Similarity Search Model (MQSSM): The Overall Approach

The models proposed herein are based on similarity searching. Multiple positive sequences are employed as queries to predict the antiviral activity of a target dataset. This one-class model identifies potential positive sequences based on their similarity to the provided references. The construction of these models involves tuning three different parameters:

Query Dataset: This is defined as the reference dataset that the model uses for predictions.

Pairwise Sequence Alignment Algorithm: Smith–Waterman (local alignment) and Needleman–Wunsch (global alignment) algorithms are employed, using the Blossum-62 substitution matrix for calculating pairwise similarity scores.

Similarity Threshold: The MQSS is a fusion model, specifically, a group fusion model. In group fusion, a similarity is computed for each reference sequence (*q*) and target sequence (*t*), denoted as *S* (*q*, *t*). All the pairwise similarity scores are combined for each sequence (*t*), and the MAX-SIM rule is applied. The similarity scores are then ranked in decreasing order. A similarity threshold (*c*) is set to determine which sequences are considered to have a positive activity. For a better understanding of this process, a graphical representation is provided in Figure 1.

### 2.2. Construction of Query/Reference Datasets

When constructing a query dataset, a critical consideration is its ability to encompass a significant portion of the antiviral active chemical space without disproportionately representing any category of AVPs or neglecting rare sequences. To achieve this, we leveraged the reductions obtained in a previous study [30].

#### Recalling the Scaffold Extraction Procedure

To ensure the comprehensive coverage of the antiviral active chemical space, we established a half-space proximal network (HSPN) using the 4663 antiviral sequences stored in StarPepDB [31], recognised as one of the most extensive databases of biologically active peptides to date (36). Several scaffold (representative AVP subset) extractions were executed using the resources available in the StarPep toolbox. During this process, we adjusted the alignment algorithm type, the pairwise identity percentage, and the centrality measures (community hub bridge [32] and harmonic [33,34]). These adjustments yielded a total of 20 scaffolds or representative AVP subsets. Subsequently, we thoroughly examined these scaffolds using a Dover Analyser [35] to assess the extent of the similarity overlap between them.

According to these observations, we selected 10 scaffolds for their representativeness and diversity. Furthermore, an additional five scaffolds were crafted by combining some of the 20 previously generated scaffolds, considering their pairwise identity percentages. These 15 scaffolds served as the foundational datasets used as references for the MQSSMs. The particularities of each of the mentioned scaffolds are briefly explained in Appendix A (Appendix A).

### 2.3. Target Datasets for Calibration and Validation of MQSSMs

We conducted a comprehensive literature review to gather a range of positive and negative sequences for testing our models. Ultimately, we selected 15 datasets, containing positive and negative sequences from synthetic and natural origins. These datasets from the literature played a crucial role in various stages of model calibration and validation. All the mentioned datasets are depicted in Table 1.

Furthermore, we created two new datasets, amassing a total of 20,136 positive sequences and 70,130 negative sequences. These 90,266 sequences underwent several filters to eliminate redundancy, resulting in an “expanded” dataset comprising 55,822 sequences, including 3178 positive and 52,644 negative sequences. The different sources for the “expanded” dataset are specified in Figure 2.

Additionally, for benchmarking against state-of-the-art methods, we curated a “reduced” dataset with specific criteria, excluding non-standard amino acids and restricting sequence lengths to between 7 and 30 amino acids, as commonly specified by many existing predictors. Refer to Figure 2 for a better depiction of the distinct filters that this dataset went through. The “reduced” dataset consists of 27,692 sequences, encompassing 1419 positive and 26,273 negative sequences. Both the “expanded and reduced” versions can be found in the Appendix A (Appendix A).

### 2.4. Construction, Selection, and Improvement of MQSSMs

As mentioned earlier, three primary parameters require fine-tuning for the construction of multi-query similarity search models (MQSSMs). The first parameter is the choice of the query dataset, for which 15 different scaffolds were evaluated. The second parameter is the selection of the alignment algorithm, alternating between global and local alignments. Lastly, the similarity threshold was varied from 0.3 to 0.9. These parameters collectively allowed for the potential creation of 210 variations of MQSSMs. Consequently, reducing the number of models was imperative.

To select the best-performing models, a two-phase approach was implemented, as depicted in Figure 3. The first phase, known as the calibration phase, aimed to significantly reduce the number of models and identify key trends in parameter calibration. The calibration phase was further divided into two rounds. In the first round, 210 models were assessed using the datasets TS_StarPep, TR_StarPep, and Ex_StarPep, resulting in a reduction to 80 models. The second round of the calibration phase involved testing the models against six different datasets: AVPIden, AMPfun, ENNAVIA-A, ENNAVIA-B, Imb, and Thakur. After this evaluation, a set of 50 models remained.

These 50 models subsequently entered the validation stage, where the primary objective was to fine-tune parameter values while also assessing the models against more specific datasets. Once again, the validation phase was subdivided into two rounds. In the first round, the 50 models were tested against five datasets: Sharma, AI4AVP, ENNAVIA-C, ENNAVIA-D, and Imb_CoV, with the last three containing anti-SARS-CoV-2 sequences. This round led to the selection of 32 models, which were further evaluated against the expanded datasets. Following this analysis, a final set of 12 models was chosen, comprising six that utilised global alignment and six that employed local alignment (M1–M12).

As 12 models still represented a considerable number, an examination was conducted to assess the degree of overlap in the sequences recovered using these different models. After a cursory examination, three base models were selected, referred to as M3, M7, and M12.

### 2.5. Scaffold Fusion

To enhance the models’ performance, scaffold fusion was performed with the aim for expanding the reference dataset. This fusion process involved combining the individual scaffolds from the best-performing base models (M3, M7, and M12) to create a new query dataset. Once this new query dataset was constructed, 14 MQSSMs were generated using it, including variations in the alignment algorithm and the similarity threshold during the creation of these models.

Subsequently, these models underwent the previously described workflow of calibration and validation. Following this process, the best-performing model was identified and is now referred to as M13 (Figure 3).

### 2.6. Scaffold Enrichment

To further enhance the base models, a query enrichment strategy was implemented. This involved gathering the positive sequences from the datasets used in the calibration phase, round 2 (Thakur, ENNAVIA-A, AMPfun, and Imb), into a single dataset. This process resulted in 2403 unique sequences. To ensure the absence of these sequences from the currently top-performing model’s scaffolds, a pairwise similarity comparison was conducted using Dover Analyser (https://mobiosd-hub.com/doveranalyzer/, v0.1.2 accessed on 24 November 2023). Subsequently, these sequences were used to construct a half-space proximal network (HSPN), similar to the one constructed for the StarPepDB AVP sequences.

With the newly curated and validated dataset in hand, it was integrated into the StarPep toolbox to construct an HSPN, similar to the HSPNs developed previously for the entire AVP space. Following HSPN construction, scaffold extraction was applied to the network, varying the centrality measure between harmonic and the community hub bridge. The alignment algorithm type was adjusted between local and global, considering only 80% and 90% as the sequence identity percentages. This process yielded 8 new scaffolds, designated as “external scaffolds”. These scaffolds were utilised to develop new MQSSMs employing the same methodology as before, resulting in 112 new models. These newly generated models underwent testing against the calibration and validation workflow of the 15 datasets to identify the most effective external scaffolds.

Finally, two models, named E1 and E2, were selected. The scaffolds from models E1 and E2 were added to those of models M3, M7, M12, and M13, ensuring that there were no repeated sequences in the enriched scaffolds. Once the enriched scaffolds were obtained, the models retained the alignment type and similarity cutoff of the base models, and they were tested against the 14 individual datasets and the expanded dataset (Figure 3).

### 2.7. Performance Evaluation

The performances of all the models and state-of-the-art models were measured using the metrics of accuracy (ACC), sensitivity (SN), specificity (SP), Mathew’s correlation coefficient (MCC), false positive rate (FPR), and F1 score as follows [41]:$$ACC=\frac{TP+TN}{TP+TN+FP+FN}$$
$$SN\;(Recall)=\frac{TP}{TP+FN}$$
$$SP=\frac{TN}{TN+FP}$$
$$\left(MCC\right)=\frac{TP\times TN-FP\times FN}{\sqrt{\left(TP+FP)\times (TP+FN)\times (TN+FP)\times (TN+FN\right)}}$$
$$Precision=\frac{TP}{TP+FP}$$
$$F1=\frac{2\times Precision\times Recall}{Precision+Recall}$$
$$FPR=\frac{FP}{FP+TN}$$
where *TP* is the true positives, *TN* is the true negatives, *FP* is the false positives, and *FN* is the false negatives.

Additionally, the Friedman test was employed to analyse the performance rankings of the various MQSS models across all the described metrics. The corresponding significance test was carried out using KEEL software (http://www.keel.es, v3.0, accessed on 1 December 2023 [42]) and the non-parametric statistical analysis module. This non-parametric test is well suited for comparing multiple models across different datasets or conditions, especially when dealing with ranked data, as in the present case.

### 2.8. Comparison with State-of-the-Art Predictors

To assess the robustness of the MQSSMs, a selection of state-of-the-art predictors was made, primarily by considering their availability through web services and ease of implementation. Table 2 provides a summary of the predictors used in this study. It is important to highlight that the use of the predictors was based on their computational availability, and some of these models show a reduced domain of application because of their training, for instance, the classifier developed to predict the antiviral actions of plant-derived peptides, PTPAMP (http://www.nipgr.ac.in/PTPAMP/, accessed on 24 August 2024). However, most of these classifiers have a general application domain because they were trained with sequences from different origins and with actions against various types of viruses. Ultimately, this experiment aims to evaluate the true merit of the predictors regardless of the learning performed. For comparison with the predictors, the reduced dataset was employed, as previously stated.

## 3. Results and Discussion

### 3.1. Performances of MQSSMs in the Calibration Phase

This segment focuses on managing the initial number of 210 models via a reduction process. In the preliminary phase of the calibration, an evaluation of these 210 models was conducted utilising three datasets: TR StarPep, TS StarPep, and EX StarPep. These datasets, predominantly derived from StarPepDB, demonstrated relatively uniform behaviours across the models’ performances, as depicted in Figure 1. Such an outcome is expected, considering the considerable overlap in the sequences shared between these datasets and the employed “query”.

During the next phase of the calibration, six datasets not associated with StarPepDB were incorporated (Figure 1). Predictably, in this phase, the models’ effectiveness was reduced compared to their effectiveness in the initial round. A critical revelation from this stage is that the models’ performance was better when dealing with datasets composed of both randomly generated negative sequences and sequences yet to be experimentally tested. In contrast, when the datasets had experimentally validated sequences, the performance metrics dropped. This behaviour might be linked to the fact that a significant number of these negative sequences closely resembled the positive ones. Consequently, the methods based on alignment faced challenges in differentiating the unique attributes of each category.

This problem was especially pronounced in datasets like ENNAVIA-A and Thakur, where experimental sequences were used as negative datasets. Conversely, with ENNAVIA-B, which contained the same positive sequences, the models more effectively identified non-antiviral sequences. It is important to emphasise that experimentally validated negative sequences are scarce in comparison to positive sequences. Hence, the predominant challenge in modelling lay in improving the detection rate (recall) of the positive sequences overall.

From the calibration stage, clear trends in the MQSSMs’ parameters emerged. A notable observation was that models with larger scaffolds performed better. This improvement was attributed to the more detailed characterisation of the AVP chemical space. Scaffolds such as Md4, Md5, SG4, SG5, SL4, and SL5 generated the most model variants (Appendix A). In terms of the alignment, the global alignment proved to be more effective with lower sequence identity percentages, while local alignments were more successful with higher percentages. For simpler scaffolds, global alignments consistently outperformed local alignments, regardless of the identity percentage.

### 3.2. Performances of MQSSMs in the Validation Phase

In the validation stage, we assessed the models against datasets with sequences targeting specific viruses, like SARS-CoV-2. Datasets including ENNAVIA-C, ENNAVIA-D, and Imb_CoV were used for this purpose. As shown in Figure 1, the base models had poor performances with ENNAVIA-C and Imb_CoV. This result was expected because the models’ references are from 2019. This highlights the importance of the “query” dataset’s representation in the model performance. However, the models showed better performances with the ENNAVIA-D dataset, which contained random negative sequences. This aligns with previous observations of the model behaviour with similar datasets.

During the model selection process, 32 models chosen in the first round of the validation were tested against the expanded dataset. From this, 12 models were identified as top performers using a multi-variable Friedman ranking method. These models, labelled from M1 to M12, included six based on global alignment and six on local alignment strategies. The parameters for these models are summarised in Appendix A. Additionally, from these 12 models, the three top-performing ones were further singled out. This selection focused efforts on fine-tuning these models, particularly towards enhancing the recovery of positive (AVP) sequences.

### 3.3. Improving MQSSM Performances by Fusing Scaffolds

Continuing from the focused analysis of the top models for positive sequence recovery, an initial approach to enhancement involved combining the scaffolds from models M3, M7, and M12 (md4, SL5, and SG5) into a single consolidated scaffold. This step included the removal of duplicate sequences, culminating in a scaffold that contained 3206 unique sequences. Subsequent testing of this modified scaffold indicated slight improvements in the performances of the models using the global alignment with a 90% similarity threshold. Labelled as M13, a new model was crafted using these parameters. The improvements in M13 were primarily because of the expanded representation of the sequence space, enriched by the increased number of sequences.

Although these modifications provided insights into the nuances of the MQSSMs, their overall efficacy remained unsatisfactory. The subsequent strategy aimed to leverage the performance information gathered during the calibration phase. As shown in Figure 1, the base models struggled with datasets such as Thakur, ENNAVIA-A, AMPfun, and AVPiden. This struggle was likely because of the inadequate representation of diverse sequences from these datasets in the MQSS scaffolds. Furthermore, the presence of many experimentally validated negative sequences in some datasets increased the difficulty of making accurate predictions.

### 3.4. Improving MQSSM Performances by Enriching the Best Scaffolds

In tackling the identified issues, a half-space proximal network (HSPN) was made by pooling together positive sequences from the challenging datasets (Thakur, ENNAVIA-A, AMPfun, and AVPiden). A total of 2403 sequences were employed in constructing the HSPN. This effort resulted in the production of eight scaffolds, out of which the top two were chosen to enhance the existing scaffolds. Consequently, this led to the introduction of new, improved models: M3+, M7+, M12+, and M13+. The “+” in their names signifies their reference enrichment. Post enhancement, these scaffolds contained 3155, 3437, 3472, and 3606 sequences, respectively. To incorporate the unique scaffolds not present in StarPepDB, we crafted E1 and E2 models using external scaffolds comprising 1517 and 1261 sequences, respectively. This increased our analysis pool to 10 models, adding six newly enriched models to the pre-existing M3, M7, M12, and M13 models. The complete details of these models are outlined in Appendix A.

Following their development, these 10 models were extensively tested across 15 databases, the 14 datasets from our workflow plus the expanded dataset (Appendix A in Appendix A). We used a Friedman ranking system to halve the number of top-performing models, evaluating them based on their accuracy (ACC), specificity (SP), sensitivity (SN), Matthew’s correlation coefficient (MCC), and F1 score. The ranking results identified M3+, M13+, M7, M12, and E1 as being the most effective, as highlighted in grey in Table 3.

Notably, although the models with a greater number of reference sequences, like M3+ and M13+, ranked high, models, such as E1, M7, and M12, with fewer sequences performed comparably well. This suggests that the effectiveness of the references hinges more on their diversity and representational range than on their quantity.

The scaffold enrichment, in general, improved the performance of the models. There were two exceptions: M12+ and M7+ showed worse accuracies (Qs) than their original models, M12 and M7, respectively. In these two cases, the increased representations of positive sequences resulted in decreases in Q because more negative sequences were identified as false positives, which is reflected in the increase in the false positive rate (FPR). In other words, in these cases, the scaffold enrichment improved the sensitivity (recall and SN) but worsened the specificity of the model.

### 3.5. Benchmarking the Best MQSSMs against State-of-the-Art Predictors

The top five models were subsequently benchmarked against existing predictors in the literature, providing a comparative assessment of their performances relative to those of other available tools. To ensure unbiased comparisons, any sequences shared between the scaffolds of models M3+, M7, M12, M13+, E1, and the reduced dataset were eliminated. This action reduced the number of positive sequences to 116, while the number of negative sequences remained unchanged. It is important to note that many negative sequences in the reduced dataset are a part of the training sets for the external predictors, but these were not removed. This decision placed our models at a significant disadvantage in terms of the performance evaluation. We tested a total of 14 external predictors (Appendix A) using evaluation metrics, such as the ACC, SP, SN, MCC, FPR, and F1 score. MCC, which is unaffected by the dataset’s imbalance, was the primary metric for our analysis.

The alteration made to the reduced dataset resulted in marked decreases in the performances of the MQSSMs. These declines were especially pronounced in the SN and MCC values. However, ACC and SP remained relatively stable, a situation due to the substantial imbalance between the positive and negative cases in the class distribution. The notable drop in the sensitivity underscores a significant consistent shortfall in correctly identifying positive sequences. This inability to accurately recall true positives notably affected the MCC, with a stark decrease observed, for example, in the M13+ model, where MCC plummeted from 0.731 to 0.214, as detailed in Table 4. The F1 score, which depends on both the recall and precision, also experienced a corresponding decline.

Despite these less-than-ideal results, the MQSSMs still surpassed the external predictors in the overall performance. Analysing the performance data of the external predictors, Figure 2 highlights two distinct trends. Some models are highly effective at identifying most positive sequences, resulting in a high SN but with the trade-off of a higher rate of false positives. Conversely, other models excel at correctly classifying all the negative sequences but tend to misclassify many positive ones, a trait observed in the MQSSMs we developed. Typically, most models based on deep learning fall into the former category, demonstrating high sensitivity, whereas traditional ML models are more likely to be in the latter category, with a stronger emphasis on specificity.

The analysis revealed that no single predictor excelled in all the evaluation categories, supporting Garcia-Jacas et al.’s conclusion that DL methods may not be the most effective for AVP prediction [36]. Extending this observation, it is evident that none of the ML models tested demonstrated satisfactory performance, suggesting a need for significant improvements. That is, the “no free lunch” theorem is fulfilled here, in the sense that there is no single (unique) best model for all the scenarios and that the use of several models is needed to improve performance and hit recovery in the general virtual screening of large databases. Therefore, these models will be more useful for general screening processes rather than for optimising lead compounds because of the structurally diverse training series that are not based on potency criteria for antiviral actions. In fact, a central issue identified is the quality and representativeness of the training data. Most positive sequences used in model training exhibit a high degree of similarity, often up to 90%. Moreover, the experimentally validated negative sequences often mirror their positive counterparts. The challenge, therefore, does not lie in the complexity of the models’ architectures but rather in the data’s availability and diversity. This highlights an ongoing challenge in gathering and effectively utilising comprehensive data for such models.

Evaluating state-of-the-art predictors revealed a common issue of accessibility. Many predictors were difficult to assess for various reasons. A significant issue was the poor construction of several web servers, leading to operational failures or frequent malfunctions. This problem affected even relatively new servers, those less than 2 years old. Furthermore, a few research repositories lacked comprehensive instructions, complicating their use. This difficulty in accessing and implementing these tools echoes concerns previously noted by researchers [46] about the availability of source codes. Nonetheless, a comprehensive summary of the prediction tools currently available is provided in the Appendix A (Appendix A).

In contrast to these challenges, the MQSSMs distinguish themselves with their accessibility. They are available through StarPep toolbox standalone software, which boasts a user-friendly interface, making these models more approachable and easier to use for users.

Despite the need for improvements to address their shortcomings, the MQSSMs still have a performance edge over traditional ML models. This is substantiated by the Friedman MCC, ACC, SP, SN, and the F1 Score. An important aspect to note is that the MQSSMs require considerably fewer computational resources and are not constrained by sequence length limitations, presenting significant benefits.

When choosing a prediction model, it is essential to align with the specific needs of the researcher. Some may prioritise identifying a larger number of potential AVPs, while others might prefer a smaller, more accurate set to reduce false positives. Considering the resource-intensive nature of the experimental procedures, the latter approach is often more practical for synthesising potential AVPs, as it optimises the balance between resource use and the likelihood of accurate predictions.

## 4. Conclusions

This research marks a significant advancement with the development of multi-query similarity search models (MQSSMs). These models, devised from a deep understanding gained in the chemical space exploration and scaffold extraction phases, employ a novel approach that leverages structural similarities for predicting biological activities. This has been instrumental in improving the model selection.

Another major achievement of this study is the assembly of the most extensive datasets of AVPs, referred to as the “expanded” and “reduced” datasets. These datasets were designed to cater to a spectrum of research needs in the fields of AVP modelling and prediction and are available as Appendix A.

Such rich data diversity is essential for the development of effective AVP predictors. Following comprehensive evaluations and filtering processes, five MQSSMs were selected for their superior performances. These models were rigorously tested across various metrics, with the top model demonstrating impressive results: ACC = 0.969, SP = 0.979, SN = 0.802, MCC = 0.71, FPR = 0.021, and F1 = 0.746. In contrast, 14 contemporary ML-based predictors, though extensively promoted, were surpassed by the MQSSMs. This not only emphasises the limitations of existing methods but also highlights the advanced capabilities of MQSSMs in predicting AVP sequences, effectively handling the challenges of variable-length sequences and imbalanced datasets. As these models notably eclipsed existing state-of-the-art predictors, they have set a new standard in the field of AVP discovery.

## Data Availability

The presented MQSSMs can be easily run using the StarPep toolbox, which is freely available at https://github.com/Grupo-Medicina-Molecular-y-Traslacional/StarPep (v0.8.5, accessed on 17 April 2023). The 4663 AVPs stored in StarPepDB can be accessed and downloaded directly through the StarPep toolbox. The half-space proximal networks (HSPNs) are also constructed using the StarPep toolbox. The datasets used in the calibration and validation stages of the MQSSMs, as well as those used for the performance comparison with existing tools, are included as a part of the Appendix A in this work.

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
