# Peer review of "Innovative Alignment-Based Method for Antiviral Peptide Prediction"

_antibiotics, 2024, doi:10.3390/antibiotics13080768_

Round 1

Reviewer 1 Report

Comments and Suggestions for Authors The MS is comprehensive to antiviral research and interesting to the general readers

Author Response

Thank you very much for your review and the positive comments on the article.

Reviewer 2 Report

Comments and Suggestions for Authors

In their work de Llano García and coworkers provided a benchmark of alignment-based method for antiviral peptide prediction. Prior to the method they also assembled the most comprehensive dataset of antiviral sequences to date. The paper is well written and the conclusions support the described analyses. However, I still have several comments to the manuscript, which can be treated as minor revision.

First, some of the descriptors selected for benchmark are very general, for example the PTPAMP predictor will be selective towards plant derived bioactive peptides and not viral peptides. The authors should make it clear in the manuscript.

If the performance drops when the dataset contains a lot of subtly different sequences (eg, experimentally validated datasets) it means that it is more useful in case of more general screening rather than a help for experimentalists who are currently optimizing a set of sequences.

The authors suggested that the most important parameter for comparing the models performance will be the detection rate (recall) of positive sequences overall (line 299). However, this parameter is not used directly in any benchmark in the manuscript.

The authors stated that enrichment of the models resulted in increased accuracy – however in two cases it did not happen. What could be the reason for that?

As a person who also tested different software I agree with the authors about the issue with accessibility and I’m also grateful for mentioning this aspect.

Finally, there is a typo in the formula for precision. Please correct it.

Author Response

Reviewer 2.

Comment 1: In their work de Llano García and coworkers provided a benchmark of alignment-based method for antiviral peptide prediction. Prior to the method they also assembled the most comprehensive dataset of antiviral sequences to date. The paper is well written, and the conclusions support the described analyses. However, I still have several comments to the manuscript, which can be treated as minor revision.

Response 1:

Thank you very much for taking the time to review this manuscript. Please find the detailed responses below,

Comment 2: First, some of the descriptors selected for benchmark are very general, for example the PTPAMP predictor will be selective towards plant derived bioactive peptides and not viral peptides. The authors should make it clear in the manuscript.

Response 2:

We agree, indeed this predictor (classifier) uses 72 antiviral peptides derived from plants. However, the three series of negative samples were extracted from other different sources (http://14.139.61.8/PTPAMP/download.php), with other origin. Taking into consideration that some amino acids in plant-derived peptides differ from those of other origins (e.g., “…the abundance of cysteine residues in plant-derived AMPs and the distribution of other residues like G, S, K, and R, which differ according to the peptide structural family”, https://link.springer.com/article/10.1007/s00726-022-03190-0). Thus, as the antiviral sequences used in the model training are ONLY of plant origin, the model's domain application is more focused on that plant-derived peptides. This fact is indicated in the paper, although part of the comparison experiment aimed to evaluate the true merit of the previously reported predictors for classifying peptides as antiviral or not. However, it is mentioned in general terms that some models have a general application domain while others are more specific, as is the case with these predictors for plant-derived peptides.

“It is important to highlight that the use of the predictors was based on their computational availability, and some of these models show a reduced domain of application due to their training. For instance, the classifier developed to predict the antiviral action of plant-derived peptides, PTPAMP (http://www.nipgr.ac.in/PTPAMP/). However, most of these classifiers have a general application domain since they were trained with sequences from different origins and with action against various types of viruses. Ultimately, this experiment aims to evaluate the true merit of the predictors regardless of the learning performed.”

Comment 3: If the performance drops when the dataset contains a lot of subtly different sequences (eg, experimentally validated datasets) it means that it is more useful in case of more general screening rather than a help for experimentalists who are currently optimizing a set of sequences.

Response 3:

OK; it is a conclusion that can be inferred from the results. Certainly, in the new version of the paper, we included a comment related to this statement. Thanks a lot!

We also focused on the fact that the "no free lunch" theorem is fulfilled in the sense that there is no single best model, and that the use of several models is needed to improve performance and hit recovery in general virtual screening of large databases. Therefore, these models will be more useful for general screening processes rather than for optimizing a lead compound, due to the structurally diverse training series that is not based on potency criteria for antiviral action.

“Extending this observation, it's evident that none of the ML models tested demonstrated satisfactory performance, suggesting a need for significant improvements. That is, the "no free lunch" theorem is fulfilled here, in the sense that there is no single (unique) best model for all scenarios, and that the use of several models is needed to improve performance and hit recovery in general virtual screening of large databases. Therefore, these models will be more useful for general screening processes rather than for optimizing a lead compound, due to the structurally diverse training series that is not based on potency criteria for antiviral action.

Comment 4: The authors suggested that the most important parameter for comparing the models performance will be the detection rate (recall) of positive sequences overall (line 299). However, this parameter is not used directly in any benchmark in the manuscript.

Response 4:

This statement appears related to a calibration/validation stage, using several specific datasets with differences in the origin of the negative sequences.

“This problem was especially pronounced in datasets like ENNAVIA-A and Thakur, where experimental sequences were used as negative datasets. Conversely, with ENNAVIA-B, which contained the same positive sequences, the models more effectively identified non-antiviral sequences. It is important to emphasize that experimentally validated negative sequences are scarce in comparison to positive sequences. Hence, the predominant challenge in modelling lay in improving the detection rate (recall) of positive sequences overall.”

Here, given the characteristics of the negative series, the purpose was as stated, BUT the quality of the models was based more on other parameters that better evaluate the performance of the models and are less sensitive to class-imbalanced data, such as the Matthews correlation coefficient. Indeed, we state in several places in the manuscript that this parameter is the most important in the evaluation of the models. For instance,

“The performance of all the models and state-of-the-art models was measure using the metrics of Accuracy (ACC), Sensitivity (SN), Specificity (SP), Mathews Correlation Coefficient (MCC), False Positive Rate (FPR) and F1 Score [41]:”

“Additionally, the Friedman test was employed to analyse the performance rankings of the various MQSS models across all described metrics”

We tested a total of 14 external predictors (File SI4), using evaluation metrics such as ACC, SP, SN, MCC, FPR and F1 Score. MCC, which is unaffected by the dataset's imbalance, was the primary metric for our analysis.

In addition, Therefore, that last sentence (Hence, the predominant challenge in modelling lay in improving the detection rate (recall) of positive sequences overall) was removed to avoid confusion.

Comment 5: The authors stated that enrichment of the models resulted in increased accuracy – however in two cases it did not happen. What could be the reason for that?

Response 5:

“The scaffold enrichment, in general, improved the performance of the models. There were two exceptions: M12+ and M7+ showed worse accuracy (Q) than their original models, M12 and M7, respectively. In these two cases, the increased representation of positive sequences resulted in a decrease in Q because more negative sequences were identified as false positives, which is reflected in the increase in the false positive rate (FPR). In other words, in these cases, the scaffold enrichment improved sensitivity (recall, SN) but worsened the specificity of the model.”

Comment 6: As a person who also tested different software I agree with the authors about the issue with accessibility and I’m also grateful for mentioning this aspect.

Response 6:

Thank you for your comment. Indeed, this is a major issue currently affecting several endpoints in this field.

Comment 7: Finally, there is a typo in the formula for precision. Please correct it.

Response 7: OK,

Reviewer 3 Report

Comments and Suggestions for Authors

The manuscript is professional, organized, and well presented.  The testing concepts are sound, obviously thought out, and logical in order.  What was not clear was the current and/or future applicability of the proposed information.  For example, does this come up with new target molecules/backbones or is it only restricted to molecular structures already discovered (i.e. the training sources)?  This entire manuscript appears to be purely theoretical, and merely an evaluation of the current methods of evaluating known results.  As the authors stated, they are restricted by the quality and quantity of the background data provided (e.g. poor websites with no documentation etc.), but what we need is a tool (or tools) to point structure/function experimental based research towards new molecules previously unrecognized or never considered.  Small variations on known molecules is of limited usefulness.  What we need is insight into the conceptual mechanisms of activity and/or new scaffolds of high probability to begin modifying.  If this was the result, it is not necessarily clear to less familiar readers.  The background content could be stronger for people not intimately knowledgeable about the field, but this may not be within the expected scope of the journal and therefore may not be practical.

Author Response

See response in attached file!
